# Qualitative and Quantitative Comparison of Plasma Exosomes from Neonates and Adults

**DOI:** 10.3390/ijms22041926

**Published:** 2021-02-15

**Authors:** Julia Peñas-Martínez, María N. Barrachina, Ernesto José Cuenca-Zamora, Ginés Luengo-Gil, Susana Belén Bravo, Eva Caparrós-Pérez, Raúl Teruel-Montoya, José Eliseo-Blanco, Vicente Vicente, Ángel García, Irene Martínez-Martínez, Francisca Ferrer-Marín

**Affiliations:** 1Servicio de Hematología y Oncología Médica, Hospital Universitario Morales Meseguer, Centro Regional de Hemodonación, Universidad de Murcia, IMIB-Arrixaca, 30003 Murcia, Spain; julia.penas@um.es (J.P.-M.); ernestojose.cuenca@um.es (E.J.C.-Z.); ginesluengo@um.es (G.L.-G.); ecp4@um.es (E.C.-P.); raulteruelmontoya@hotmail.com (R.T.-M.); vicente.vicente@carm.es (V.V.); 2Platelet Proteomics Group, Centro Singular de Investigación en Medicina Molecular y Enfermedades Crónicas (CIMUS), Universidad de Santiago de Compostela e Instituto de Investigación Sanitaria de Santiago (IDIS), 15706 Santiago de Compostela, Spain; Maria.Barrachina@childrens.harvard.edu (M.N.B.); angel.garcia@usc.es (Á.G.); 3Grupo de Investigación en Patología Molecular y Farmacogenética, Departamento de Dermatología, Estomatología, Radiología y Medicina Física, Hospital General Universitario Santa Lucía, Universidad de Murcia, IMIB-Arrixaca, 30202 Cartagena, Spain; 4Servicio de Proteomica, e Instituto de Investigación Sanitaria de Santiago de Compostela (IDIS), Hospital ClínicoUniversitario de Santiago de Compostela, 15706 Santiago de Compostela, Spain; sbbravo@gmail.com; 5U-765-CIBERER, Instituto de Salud Carlos III (ISCIII), 28220 Madrid, Spain; 6Servicio de Obstetricia y Ginecología, Hospital Clínico Virgen de la Arrixaca, 30120 Murcia, Spain; jeblancoc@gmail.com; 7Grado de Medicina, Universidad Católica San Antonio de Murcia, 30107 Murcia, Spain

**Keywords:** exosomes, neonatal platelets, platelet transfusion, protein S, proteomic, von Willebrand factor

## Abstract

Exosomes are extracellular vesicles that contain nucleic acids, lipids and metabolites, and play a critical role in health and disease as mediators of intercellular communication. The majority of extracellular vesicles in the blood are platelet-derived. Compared to adults, neonatal platelets are hyporeactive and show impaired granule release, associated with defects in Soluble N-ethylmaleimide-sensitive fusion Attachment protein REceptor (SNARE) proteins. Since these proteins participate in biogenesis of exosomes, we investigated the potential differences between newborn and adult plasma-derived exosomes. Plasma-derived exosomes were isolated by ultracentrifugation of umbilical cord blood from full-term neonates or peripheral blood from adults. Exosome characterization included size determination by transmission electron microscopy and quantitative proteomic analysis. Plasma-derived exosomes from neonates were significantly smaller and contained 65% less protein than those from adults. Remarkably, 131 proteins were found to be differentially expressed, 83 overexpressed and 48 underexpressed in neonatal (vs. adult) exosomes. Whereas the upregulated proteins in plasma exosomes from neonates are associated with platelet activation, coagulation and granule secretion, most of the underexpressed proteins are immunoglobulins. This is the first study showing that exosome size and content change with age. Our findings may contribute to elucidating the potential “developmental hemostatic mismatch risk” associated with transfusions containing plasma exosomes from adults.

## 1. Introduction

It is widely accepted that the number (150,000–450,000/µL) and structure of platelets in healthy neonates closely resembles that of adult platelets. However, platelet reactivity in response to agonists is reduced in this infantile age group [1,2,3,4,5]. The first evidence of developmental differences between neonatal and adult platelets came from platelet aggregation studies in full-term cord blood (CB)-derived platelet-rich plasma. New technologies using smaller blood volumes have allowed, however, the study of platelet function in whole blood, both in CB and in peripheral neonatal blood. As closure times measured by the platelet function analyzer (PFA)-100 are longer in neonatal blood compared to CB samples [6], some authors have suggested that cord platelets are functionally distinct to peripheral neonatal platelets. Further studies have shown, however, that neonatal platelets are hyporeactive independently of the source of blood (CB or neonatal peripheral blood) [7].

Beyond its role in hemostasis and thrombosis, platelets also play critical roles in other physiological and pathological processes, including inflammation, immune response or cancer [8]. Therefore, changes during development affect not only platelet reactivity, but many other aspects of platelet biology as well [9,10,11,12]. Whether and how these ontogenetic differences influence their effect “out-of target” and in the interplay between thrombosis, inflammation and immunity during development is poorly known.

Despite the poor reactivity of platelets during the fetal/neonatal life, compared with older children or adults, healthy full-term infants exhibit normal to increased primary hemostasis due to factors in neonatal blood that enhance the platelet vessel wall interaction (i.e., increased von Willebrand factor (vWF) levels and function, higher hematocrit levels or higher mean corpuscular volumes of erythrocytes) [1]. Thus, platelet hyporeactiveness during the first weeks of life is seen as part of an exceptional and well-balanced haemostatic system [1,13].

Besides the hyporeactivity, fetal and neonatal platelets present secretion defects. A recent study showed decreased dense granules but a comparable number of alpha-granules [14]. Since the number and content of α-granules, which are the most abundant, are similar between neonates and adults [15], immature signal transduction pathways have been traditionally postulated as the mechanism to explain the defective granule exocytosis of neonatal platelets [4]. Our group has recently demonstrated that the expression levels of the main SNARE (Soluble N-ethylmaleimide-sensitive fusion Attachment protein (SNAP) REceptor) proteins, which have a crucial role in the exocytosis of secretory cells [16], are developmentally regulated [15]. SNARE proteins mediate the membrane fusion process [17] between the plasma/target (t) membrane and the granular/vesicle (v) membrane [16,18,19,20]. Specifically, in platelets, the fusogenic complex is composed of one v-SNARE vesicle-associated membrane protein 8 (VAMP8) and two t-SNAREs (syntaxin-11 and SNAP-23). Interactions between v-and t-SNAREs are controlled by regulator proteins, the Sec 1/Munc proteins [17]. Compared to adult platelets, neonatal platelets showed significantly reduced levels of syntaxin-11 and Munc18 [15,16,21,22,23].

SNARE proteins also participate in the membrane fusion process that allows the biogenesis and secretion of exosomes [16,18,19,20,24]. Since the majority of extracellular vesicles in the blood (60 to 90%), including exosomes, are platelet-derived [25], in the present study we hypothesized that defects found in the neonatal platelet SNARE machine could have effects on the content or size of the plasma exosomes in newborns compared to adults.

In the present study, we provide the first comprehensive comparison of neonate and adult plasma exosomes. Using ultrastructural microscopy and mass spectrometry techniques on umbilical CB and adult peripheral blood, we performed the first proteomic analysis of exosomes throughout human development. 

## 2. Results

### 2.1. Characterization of Plasma Exosomes from Neonates and Adults

Representation of the sequential steps following blood extraction: exosome isolation, characterization and analysis of results are represented as a workflow diagram in Figure 1. Although for adult samples, plasma was isolated from peripheral blood, the large amount of blood needed for the experiments precluded this option for neonate samples. Thus, plasma from neonates was isolated from umbilical CB. Then, exosomes were isolated from plasma of adults and neonates. In order to confirm that isolated exosomes from CB were representative of neonatal plasma without contamination from the placenta, we measured the expression of miR-141. This miRNA has been reported as the miRNA with the highest expression in placenta. In fact, miR-141 expression levels are >10-fold higher in placenta than in maternal blood, and they become undetectable in postdelivery maternal plasma [26]. We did not detect expression of miR-141, either in exosomes from CB or in those purified from adult peripheral blood. As a control, besides checking the expression of this miRNA in the exosomes, we used a placenta specimen, in which we detected miR-141 expression with a Ct value of 26.1 being the miR-16-5p Ct value (used as endogenous control miRNA) of 21.8 in placenta, and 29.7 ± 0.2 and 26.2 ± 0.8 in exosomes from adult and neonate, respectively.

Besides checking the absence of placenta material in exosomes isolated from CB, the quality of the samples was characterized by dynamic light scattering analysis (DLS). As expected, we observed that the majority of exosomes had a size distribution between 10 and 100 nm (Appendix A).

Next, we evaluated the total protein content in exosomes from both age groups. Plasma exosomes from neonates presented 65% less of total protein than those from adults (0.94 ± 0.36 vs. 0.32 ± 0.17 mg/mL, *p* = 0.05). In line with the lower amount of total proteins, plasma exosomes from neonates were clearly smaller than those from adults, as determined by transmission electron microscopy (TEM, Figure 2A). The quantification of these differences showed significant variations in size in two of the three ranges of diameter studied (Figure 2B). Whereas neonatal samples were enriched with exosomes inside the 30–50 nm diameter range (neonate: 92.10 ± 2.92% vs. adult: 69.61 ± 10.75%, *p* < 0.05), adult samples showed 29.8% and nearly 10% more exosomes with a size between 50 and 80 nm (*p* < 0.05) and 81–120 nm, respectively, compared to neonatal samples (Figure 2B).

### 2.2. Differential Protein Expression

Focusing on the differential proteome study, we identified by sequential window acquisition that of all theoretical mass spectrometry (SWATH MS), 131 proteins were differentially expressed (*p* < 0.05): 83 proteins were overexpressed whereas 48 were infraexpressed in neonatal exosomes compared to adult exosomes (Appendix A). To identify the proteins with the greatest expression differences, we also filtered by Fold Change (FCh) > 2 or <0.5 and identified 104 differentially expressed proteins, 64 of which were upregulated and 40 were downregulated in neonatal exosomes (Table 1).

The unsupervised hierarchical clustering analysis (heat map) demonstrated a clear discrimination between the two groups of samples (neonates and adults) (Figure 3A). Furthermore, a principal component analysis (PCA), which is another unsupervised method, also recognized two groups of samples that represented the two developmental states (Figure 3B).

Focusing attention on the upregulated proteins in plasma exosomes from neonates, we noticed that they could be divided into two main groups. The first group included proteins that are important in the maintenance of hemostasis and participate in platelet activation, blood coagulation and thrombus formation (e.g., platelet factor 4 (PF4); talin-1 (TLN1); vWF, coagulation factor VIII (FVIII) and factor V (FV); fibrinogen, α2-macroglobulin (A2MG); Ras-related protein Rap-1A (RAP1A)) (Table 1, Figure 3C). The second group identified proteins that play a central role in exocytosis and are involved in granule secretion (e.g., Ras-related protein Rab-11B (RAB11B); clathrin; syntaxin-7; metalloreductase STEAP3; heat shock cognate 71 kDa) (Table 1, Appendix A). Interestingly, some of the differentially regulated proteins identified play a central role in the activation of the classical pathway of the complement system and are involved in inflammatory response (e.g., complement C3, C7 and C4b). Moreover, among the top 10 upregulated proteins in neonatal exosomes, three are related with erythroid-derived proteins (transferrin receptor 1; hemoglobin subunit γ-2; ferritin light chain) and two of them with proteins highly expressed in platelets (PF4 and platelet glycoprotein 4 or CD36) (Table 1, Appendix A). 

As far as the most underexpressed proteins in neonatal exosomes are concerned, most of them (24 out of 40) are immunoglobulins (Table 1). In addition, we highlight the significant downregulation of protein S (FCh of 0.42 vs. exosomes from adults) (Table 1), a vitamin K-dependent anticoagulant protein, that acts as a cofactor to protein C in the inactivation of FVa and FVIIIa.

### 2.3. Biological Pathway Analysis

The functional enrichment and interaction network analysis of neonate vs. adult proteomes showed differences in biological processes and pathways related to hemostasis, formation of fibrin clot, integrin cell surface interactions, complement cascade, β3 integrin cell surface interactions or platelet activation, signaling and aggregation (*p* < 0.001) (Table 2). By contrast, the interaction analysis performed using proteins downregulated in neonates vs. adults showed differences in biological processes and pathways related to chylomicron-mediated lipid transport and lipoprotein metabolism (*p* = 0.006 and *p* = 0.03, respectively).

In line with the previous analysis, the pathway analysis using Reactome revealed pathways that contain proteins expressed differently in exosomes from adults and neonates that are of great interest, such as developmental biology; vesicle-mediated transport; and transport of small molecules or cell-cell communication (Appendix A). One of them involves hemostasis, which includes proteins differentially expressed, related to platelet reactivity; interaction with the endothelial wall; and megakaryopoiesis (Figure 4A): vWF, coagulation factors V, VIII and XIII, PF4, RAP1A, TLN1, integrin β3 or filamin-A. In Figure 4B, those proteins expressed differently in these two age groups and that participate in vesicle-mediated transport are shown (Figure 4B): clathrin, apolipoproteins B-100 and L1, α-1B tubulin and RAB11B are found in this pathway.

### 2.4. Validation of Differential Protein Expression in Independent Samples

According to the FunRich analysis results, we decided to validate two proteins, one identified as upregulated and another identified as downregulated protein in neonates vs. adults. The protein selected among those upregulated in neonates was vWF, since this protein has been reported to be augmented in neonatal plasma as a compensatory mechanism of the relative platelet hyporeactivity of neonates [1]. In the case of proteins downregulated in neonates, we decided to validate the protein S expression, since this protein is an important natural anticoagulant whose deficiency may increase the thrombotic risk [27]. We used an enzyme-linked immunosorbent assay (ELISA) kit to measure vWF levels in 10 independent exosome samples from both groups (*n* = 10/group). Total vWF concentration in neonate plasmatic exosomes was significantly higher than in adult plasmatic exosomes (14.78 ± 6.17% vs. 6.17 ± 2.77%; *p* ≤ 0.001) (Figure 5A). Protein S levels were validated in four independent exosome samples from both groups by Western blotting, using a specific primary antibody against protein S and the detection of tumor susceptibility gene 101 (TSG-101) protein as an exosome-specific marker. Total protein S concentration in neonate plasmatic exosomes was lower than in adult plasmatic exosomes (0.47 ± 0.30 and 1.23 ± 0.78, respectively; *p* = 0.05) (Figure 5B).

## 3. Discussion

Exosomes are spherical extracellular vesicles with a diameter between 30 and 160 nm (average ~100 nm). Extracellular vesicles also include apoptotic bodies and microvesicles, which differ from exosomes by their size (apoptotic bodies: 1–5 µm; microvesicles or microparticles: 0.1–1 µm; exosomes: 30–160 nm) for having specific markers and different biogenesis. Whereas microparticles and apoptotic bodies are generated by plasmatic membrane gemmation, the exosomes come from multivesicular bodies by the endocytic pathway [17,28,29]. In this context, the exosome content, which includes different proteins, lipids and nucleic acids, could derive both from the secretory cell and from the extracellular medium. Therefore, the exosome composition seems to be a reflection of the physiological status of the secretory cell [30,31]. 

Although exosomes are released by many cell types, those contained in the plasma are mainly, but not only, platelet-derived, as determined by the expression of the CD41 and CD63 markers [25,32]. Since (i) these nanosized particles are considered to be minimaps of their cells of origin [33]; (ii) the formation and secretion of exosomes is regulated by Rab and SNARE proteins [34,35]; and (iii) variations in platelet transcriptome [36], signaling pathways [37] and SNARE proteins [15] have been recently found during development, in the present study, we examined whether the morphology and/or the content of plasma exosomes from neonates was different from those of adults.

Surprisingly, we found that neonatal exosomes were smaller than those of adults. The reasons for this smaller size are unclear, but it is important to recognize that both in the formation and in the process of multivesicular bodies (MVBs), docking and subsequent fusion with the plasma membrane is regulated by Rab, SNARE and Sec 1 proteins [34]. Specifically, Sec 1 family domain containing 2 (SCFD2), Rab 31 and VAMP7 have been implicated in the fusion event between MVBs at the plasma membrane [35]. Whether the underexpression of *VAMP7*, *SCFD2*, or *Rab 31* mRNA found in neonatal (compared to adult) platelets [7,36] may contribute to the lower size of plasma exosomes from neonates needs further investigation. 

According to its smaller size, the protein concentration of plasma neonatal exosomes was also lower. Since the functional consequences of these observations were unknown, we decided to study quantitative differences in the protein content of exosomes from the two age groups. First, we ruled out possible contamination by other extracellular vesicles in our samples. Indeed, DLS analysis confirmed that the samples were enriched with particles between 10 and 100 nm in diameter, which is within the size of exosomes [38]. Then, we carried out an MS-based proteomic analysis. SWATH MS analysis showed 104 differently expressed proteins (*p*-value < 0.05, FCh > 2) between plasma exosomes from neonates and adults: 64 overexpressed in exosomes from neonates and 40 underexpressed. Regarding the latter, probably as a result of an incomplete and emerging immune system, since they have not been exposed to the environment as much as adults [39], more than 50% of the proteins underexpressed in neonatal exosomes were immunoglobulins (e.g., immunoglobulin heavy constant α 1: FCh = 0.02). 

Regarding overexpressed proteins, newborn plasma exosomes exhibited an abundance of proteins involved in platelet function and primary hemostasis, such as platelet activation and signaling proteins (integrins αIIb and β3; guanine nucleotide-binding protein G(i) subunit α-2 (GNAI2), filamin-A; TLN1, RAP1A, CD9); ligand receptor (CD36); platelet chemoattractant proteins (PF4); structural and cytoskeletal proteins (tubulin α-1B chain, β-actin, TLN1, filamin-A). Interestingly, PF4 may exert a proinflammatory effect, due to its chemotactic and neutrophil-activating effect besides its well-known role in hemostasis [40]. Worthy of special note is GNAI2, the main member of the Gαi family expressed on platelets [41]. In fact, in line with the higher levels of GNAI2 in plasma exosomes from neonates, we previously found that the expression levels of the gene that encode this protein, GNAI2, are significantly higher in neonatal platelets than in adult platelets [13]. 

Another extensive group of overexpressed protein in CB exosomes is related to the clotting cascade (A2MG, FV, FVIII, vWF, fibrinogen α, β and γ chain precursor), suggesting that exosomes may play a role in hemostatic balance during development. Interestingly, the plasma concentrations of fibrinogen, FV, FVIII and vWF are not decreased at birth [42]. Indeed, as in CB exosomes, vWF and A2MG levels are higher in plasma from fetus and neonates compared to adults [42,43]. Another group of differently expressed proteins is related to intracellular trafficking and endocytosis (e.g., RAB11B; clathrin; syntaxin-7; metalloreductase STEAP3; heat shock cognate 71 kDa). These findings supported and expanded our recent observation that, in platelets, some SNARE proteins are developmentally regulated [15,36]. We also observed positive regulation of the expression of major proteins related to integrin αIIbβ3 signaling (RAP1A; integrins αIIb and β3; TLN1), suggesting that they may contribute to balance the defective activation of integrin αIIbβ3 in platelets of preterm and full-term newborns [37]. 

Among the proteins differentially expressed in plasma exosomes throughout development, we focused on two related to the hemostatic system for further validation: vWF and protein S, a pro- and anticoagulant protein, respectively. In contrast to the higher plasma levels of vWF [42], the plasma concentration of protein S is reduced at birth by about 40% and reaches adult values by approximately three months of age. Consistent with our proteomic findings, and probably as a reflection of the plasma levels during fetal/neonatal life [42], our validation studies confirmed that vWF levels were found higher in plasma exosomes from neonates, whereas protein S levels were decreased compared to exosomes from adults. Overall, our data show that plasma exosomes from neonates are richer in procoagulant than in anticoagulant proteins. Our findings are in agreement with previous studies showing significantly increased procoagulant activity of microparticles from newborn cord plasma as compared to adult plasma [44,45]. 

Our current results are of potential clinical significance. Thrombocytopenic preterm neonates are transfused with platelets from adult donors to prevent bleeding. In the recently published Platelets for Neonatal Transfusion-Study 2 (PlaNeT-2), Stanworth et al. showed that, among preterm infants with severe thrombocytopenia, the use of a platelet count threshold of 50,000 per mm^3^ for prophylactic platelet transfusion resulted in a higher rate of death, major bleeding and bronchopulmonary dysplasia than a restrictive threshold of 25,000 per mm^3^ [46]. Although the mechanisms mediating these adverse effects are unknown, authors speculate about the inflammatory consequences after transfusion of platelet components as biologic agents (platelet-derived reactive oxygen species, proangiogenic factors or platelet transfusion-derived bioreactive components) [46]. Interestingly, at the time of the PlaNeT2 study (2011–2017), leuko-reduced platelet apheresis that thrombopenic neonates received were diluted in plasma 100%. We speculated that the presence of exosomes from adult donors in platelet apheresis (which are less coagulant than infant exosomes) can tilt neonatal hemostatic balance toward a state that promotes bleeding. Previous studies have communicated the potential “developmental hemostatic mismatch risk” associated with platelet transfusions [47].

There are several limitations of this study. First, our measurements were done in plasma, derived from umbilical CB, which does not exactly represent the hemostatic environment in the newborn. However, since maternal coagulation factors cannot cross the placenta, our findings related to the pro- and anticoagulant proteins seem to be more a reflection of global hemostasis status at birth than of contamination from other tissues or cells. Secondly, in our study, exosomes were isolated by an ultracentrifugation-based method, which remains the gold standard, but we cannot discard minor contamination by other extracellular vesicles. Actually, the problem of the non-existence, to date, of a standardized method that is effective for isolating exosomes without errors is something that the committee for the standardization and rigor of extracellular vesicles studies is trying to put on paper [48]. Third, we determined the protein content by antigen, without assessing the activity. Finally, we recognize the small sample size of the two age groups studied.

In conclusion, to the best of our knowledge, this is the first study showing that exosome formation, size and content change with age, i.e., during human development. Additionally, some of the diseases that manifest in adulthood, where exosomes may play a role, such as metabolic and cardiovascular diseases, neurodegeneration and cancer [38], are thought to start in childhood. Exosomes currently have the potential to become the fundamental approach of liquid biopsy, especially in the tumor field [49]. Consequently, there is the need to investigate exosomal diversity starting from the first years of life of patients—at an early stage—it is an excellent approach to lay the foundations for more precise analyses. Therefore, research about exosome generation and content change with age can provide information into programmed aging and how these nanovesicles can participate in the pathogenesis of these diseases.

## 4. Materials and Methods

### 4.1. Sample Collection

The study was approved by the institutional review board of Virgen de la Arrixaca Clinical Hospital. Written informed consent was obtained from all subjects in accordance with the Declaration of Helsinki. Extraction from blood of umbilical cords of healthy full-term neonates (*n* = 14) was performed at a gestational age of 38–41 weeks. The exclusion criteria for sample collection were women in labor who had taken antibiotics or anti-inflammatory drugs 10 days before birth, family medical history of thrombocytopenia, platelet function disorder, evidence of infections, gestational diabetes, preeclampsia, coagulation disorders, delayed uterine involution and substance or alcohol abuse. Concurrently, extraction from healthy adult peripheral blood venipuncture (*n* = 12) with no antiplatelet treatment 10 days before the extraction, was drawn. All samples were collected in 3.2% sodium citrate tubes (0.105 M buffered citrate, Diagnostica Stago Becton Dickinson, Plymouth, United Kingdom).

### 4.2. Platelet-Free Plasma Isolation

Blood samples were acidified with 10% Acid citrate dextrose solution (ACD), pH 4.5 (97 mM sodium citrate, 111 mM glucose, 78 mM citric acid), and centrifuged at 2500 g for 15 min to platelet-poor plasma extraction. Then, prostacyclin 1 µM (PGI2, 18220, Cayman Chemical Company, Ann Arbor, MI, USA) was added and a second centrifugation was carried out at 2500 g for 10 min to recover the plasma supernatant.

### 4.3. Plasma Exosomes Isolation and Characterization

Plasma was diluted in 4-(2-hydroxyethyl)-1-piperazineethanesulfonic acid (HEPES) buffer (1:1.4 dilution) and centrifuged at 2000× *g* for 1 h at 4 °C, to discard microvesicles. The supernatant was then ultracentrifuged at 110,000× *g* for 18 h at 4 °C and, the resultant pellet was resuspended in 2 mL KBr (0.25 M) and incubated in ice for 20 min. A sequential ultracentrifugation was carried out at 110,000× *g* for 1 h at 4 °C to pellet exosomes, which were then washed by suspension in 1 mL phosphate-buffered saline (PBS) to eliminate possible residual KBr. Finally, these vesicles where ultracentrifuged at 110,000× *g* for 1 h at 4 °C to obtain an exosomal pellet which was resuspended in 50 μL PBS. An aliquot was taken to determine protein concentration using the Pierce Coomassie Plus (Bradford) kit (Thermo Scientific, Madrid, Spain). Exosomes were stored at −80 °C.

In order to confirm that isolated exosomes from umbilical CB were representative of neonatal blood, and were not contaminated by the placenta, we measured the expression of a placental-highly expressed miRNA, miR-141 [26] and miR-16-5p as endogenous control. MiRNAs from exosomes were purified as previously described [30]. cDNA was synthesized from 50 ng RNA using individual miRNA-specific retrotranscription primers contained in the TaqMan^®^ MiRNA Reverse Transcription Kit (Applied Biosystems, Madrid, Spain). Each cDNA was amplified using the TaqMan^®^ MiRNA assays together with the TaqMan^®^ Universal PCR Master Mix, No AmpErase^®^ UNG (Applied Biosystems, Madrid, Spain). We employed the 2^−ΔCt^ method to calculate the relative abundance of miRNAs compared with miR-16-5p expression.

For the DLS, exosomes were normally isolated and suspended in 1× PBS to obtain the volume distribution of the sample. The area under each peak was integrated to produce an estimation of the percentage of volume of each population of vesicles (1:10 dilution). For this analysis, a Nanosizer ZS (Malvern Products, Madrid, Spain) was used.

Finally, five µL of exosome suspension was plated on a Formvar carbon rack (Aname, Madrid, Spain) for 5 min at 22 °C. Samples were then washed (1 drop of H_2_O Milli-Q, 1 min), fixed in 2.5% glutaraldehyde (1 min), and after a new wash, treated with 2% uranyl acetate as contrast agent. TEM images were obtained using a JEOL JEM 1011 TEM (JEOL USA Inc., MA, USA), operating at an acceleration voltage of 100 kV. The samples were observed at 100,000× magnification. The images were processed with MIP4 Advanced software (MIP4 Advanced System; Microm, Barcelona, Spain), which allowed us to determine exosome diameter. For morphometric analysis, 40 images were obtained for each of the neonate and adult samples (*n* = 3/group). Quantitative analysis of exosome diameters was performed using Fiji ImageJ software.

### 4.4. Quantitative Proteomic Studies by Liquid Chromatography Mass Spectrometry (LC-MS/MS) Using Sequential Window Acquisition of All Theoretical Mass Spectrometry (SWATH MS) Method

#### 4.4.1. Protein Digestion

Protein extract (100 μg) from adults and newborn plasma exosomes (*n* = 6/group) was loaded on a 10% SDS-PAGE gel to initiate whole protein concentration. The gel was stained and the band was exscinded and submitted to an in-gel tryptic digestion [50]. Peptides were extracted by carrying out three 20 min incubations in 40 μL of 60% acetonitrile dissolved in 0.5% HCOOH, then pooled, concentrated (SpeedVac, Thermo Fisher Scientific, Madrid, Spain) and stored at −2 °C.

#### 4.4.2. Mass Spectrometric Analysis by Sequential Window Acquisition of All Theoretical Mass Spectra (SWATH MS)

SWATH MS acquisition was performed on a TripleTOF^®^ 6600 LC-MS/MS system (AB SCIEX) at Proteomics Unit (Instituto de Investigación Sanitaria de Santiago de Compostela (IDIS), Spain). All peptide solutions were analyzed by a shotgun data-dependent acquisition (DDA) approach by micro-LC-MS/MS, an analysis already described in the literature [51,52,53]. Therefore, ion density found in the DDA runs was used to create the windows necessary in the SWATH MS method. In order to construct the MS/MS spectral libraries, adult and newborn (*n* = 6/group) were equally pooled in 3 samples. Therefore, for each developmental stage, 3 pool samples, each consisting of pooled plasma exosomes from 2 newborns or 2 adults, were run as described previously in the literature [52,53,54,55,56,57,58]. Samples were analyzed using a data-independent acquisition method, making 3 technical replicates for each sample (18 total samples: 3 replicates/sample and 3 samples/group). The targeted data extraction from the SWATH MS runs was performed by PeakView v.2.2 (ABSciex, USA) using the SWATH MS Acquisition MicroApp v.2.0 (ABSciex, USA) and the data were processed using the spectral library created from DDA. SWATH MS quantization was attempted for all proteins in the ion library that were identified by ProteinPilot with a false discovery rate (FDR) below 1%. PeakView computed an FDR and a score for each assigned peptide according to the chromatographic and spectra components; only peptides with an FDR below 1%, 10 peptides and 7 transitions per peptide, were used for protein quantization.

The integrated peak areas were processed by MarkerView software version (AB SCIEX, USA) for a data-independent method for relative quantitative analysis [52,53,54,55,56,57,58,59,60,61,62]. To control for possible uneven sample loss across the different samples during the sample preparation process, we performed a most likely ratio normalization [54,57]. Unsupervised multivariate statistical analysis using PCA was performed to compare the data across the samples. A Student’s *t*-test analysis on the averaged area sums of all the transitions derived for each protein in every sample will indicate how well each variable distinguishes the two groups, reported as a *p*-value. For each library, its set of differentially expressed proteins (*p*-value < 0.05) with a FCh > 2 or <0.5 was selected.

### 4.5. Functional Enrichment and Interaction Network Analysis

We performed a functional enrichment and interaction network analysis of neonate vs. adult proteomes using FunRich: Functional Enrichment analysis tool (http://funrich.org/ (accessed on 25 January 2021)) [63,64]. In a first step, differently expressed proteins between neonates and adult exosomes were filtered by FCh > 1.5 and *p* value (<0.05). Then, the complete lists of proteins identified in the FunRich database were entered for processing in the FunRich software. FunRich implements hypergeometric tests, FDR (Benjamini–Hochberg procedure) and the Bonferroni correction. The complete lists of proteins filtered for the analysis in FunRich are available in Table 1.

We conducted a pathway analysis using Reactome (https://reactome.org (accessed on 25 January 2021)), which uses a statistical (hypergeometric distribution) test that determines whether certain pathways are over-represented (enriched) in the submitted data and produces a probability score, which is corrected for FDR using the Benjamini–Hochberg method. Most significant pathways were represented using Reactome pathway diagrams. 

Volcano plots were performed using GraphPad Prism (GraphPad Software, San Diego, CA, USA) and a heat map was carried out using http://www.heatmapper.ca/expression (accessed on 25 January 2021).

### 4.6. Von Willebrand Factor and Protein S Determination

An ELISA kit to measure vWF levels in 10 independent exosome samples from both groups (adults and neonates) was purchased from Thermo Scientific (Madrid, Spain) (catalog #EHvWF) and used according to the manufacturer’s instructions.

Protein S was validated in 4 individual exosome samples from each group. Samples were separated on 8% SDS-PAGE, electrotransferred to a polyvinylidene difluoride membrane (GE Healthcare, Barcelona, Spain) and blocked with 10% whole milk in TBS-T (Tris-HCl, NaCl, Tween 20) for 30 min at room temperature. The membranes were immunoblotted with primary antibody protein S (dilution 1:5000) (kindly provided by Dr. F. España from Hospital Universitario y Politécnico La Fe, Valencia, Spain) at 4 °C overnight, and then with adequate horseradish peroxidase-coupled secondary antibody from Sigma (GE Healthcare, Barcelona, Spain). The chemiluminescence signals were collected using an LAS 4000 system (ImageQuant LAS 4000 mini, General Electric Company, Boston, MA, USA), and the quantitative analysis was performed using ImageJ software.

### 4.7. Statistical Analysis

All analyses were performed using SPSS Statistics 21.0 software (IBM, Armonk, NY, USA). A one- or two-tailed Student’s *t*-test or a Mann–Whitney U test was used, as appropriate. The statistical significance was defined as *p* < 0.05.

## Figures and Tables

**Figure 1 ijms-22-01926-f001:**
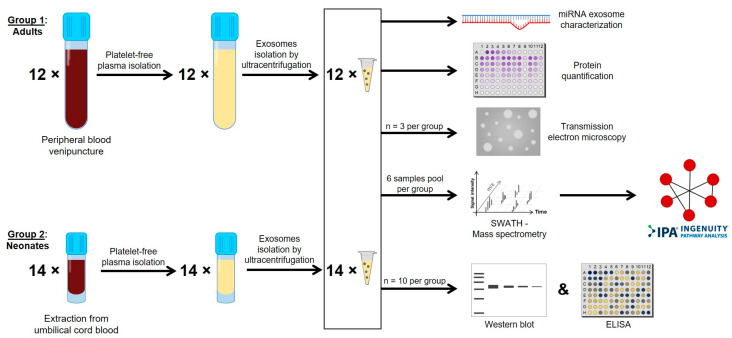
Workflow diagram. Representation of the sequential steps followed blood extraction to exosomes isolation and characterization, and analysis of results.

**Figure 2 ijms-22-01926-f002:**
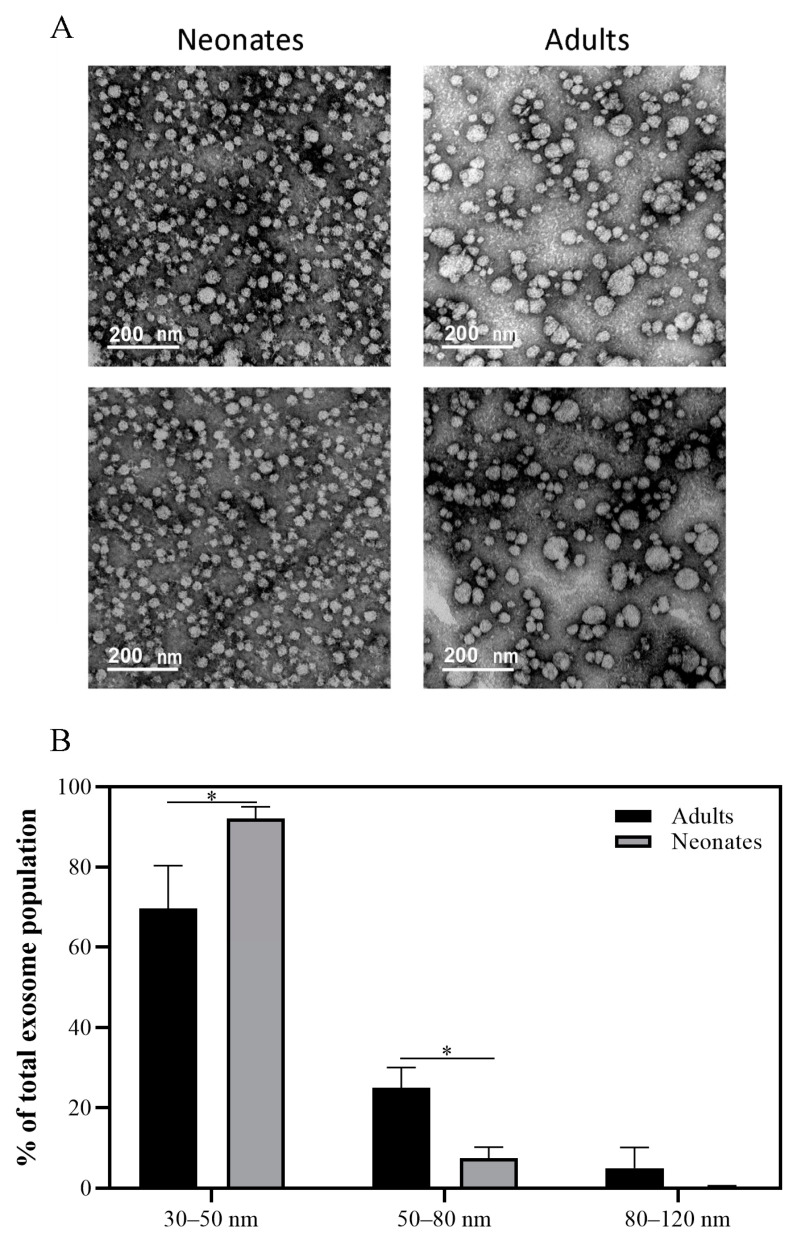
Characterization of exosomes. (**A**) Representative images of plasmatic adult and neonate exosomes from the transmission electron microscopy (TEM) (*n* = 3/group). The scale bars of images are 200 nm. The acquisition was at 100 kV. The magnification was 100,000×. The images were processed using MIP4 software. (**B**) Percentage of exosomes in different diameter size ranges (30–50 nm; 50–80 nm; 80–120 nm) in adult and neonate samples. Values represent average population percentage relative to total exosome population ± SD (*n* = 3/group); (* *p* < 0.05).

**Figure 3 ijms-22-01926-f003:**
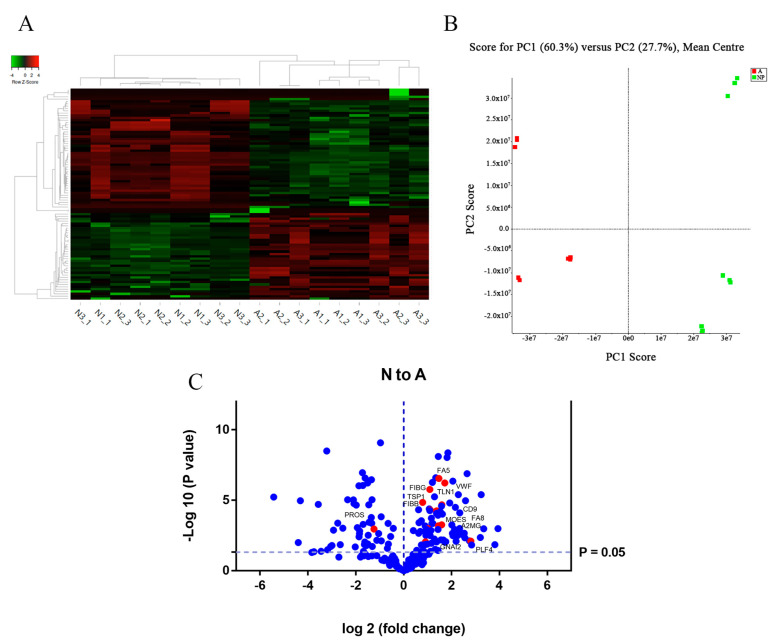
Unsupervised analysis of differentially expressed proteins in plasma exosomes from neonates vs. adults by SWATH MS. (**A**) Heat map showing hierarchical clustering between adult and neonate exosomes using top 50 differentially expressed proteins. Analysis was carried out in 3 pool samples, each consisting of pooled plasma exosomes from 2 newborns or 2 adults. (**B**) PCA analysis showing separation of adult (red) and neonate (green) exosome samples. (**C**) Volcano diagram resulting from comparison of newborn and adult exosomes. Proteins are separated according to the log_2_ of the FCh (*x*-axis) and the −log_10_ of the *p*-values based on a two-tailed *t*-test (*y*-axis).

**Figure 4 ijms-22-01926-f004:**
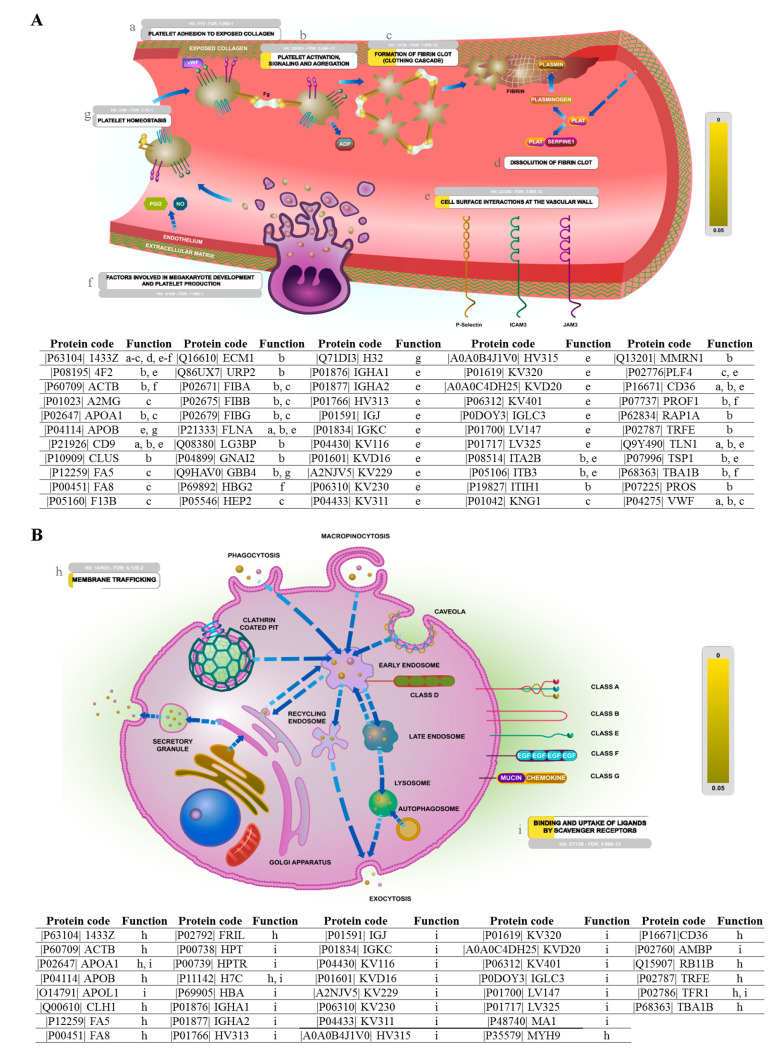
Reactome network view of differentially expressed proteins. (**A**) Hemostasis pathway showing differentially expressed proteins in adults and neonatal exosomes involved in different subpathways (a–g). (**B**) Vesicle-mediated transport pathway showing differentially expressed proteins in adults and neonatal exosomes involved in membrane trafficking (h) or binding and uptake of ligands by scavenger receptors (i).

**Figure 5 ijms-22-01926-f005:**
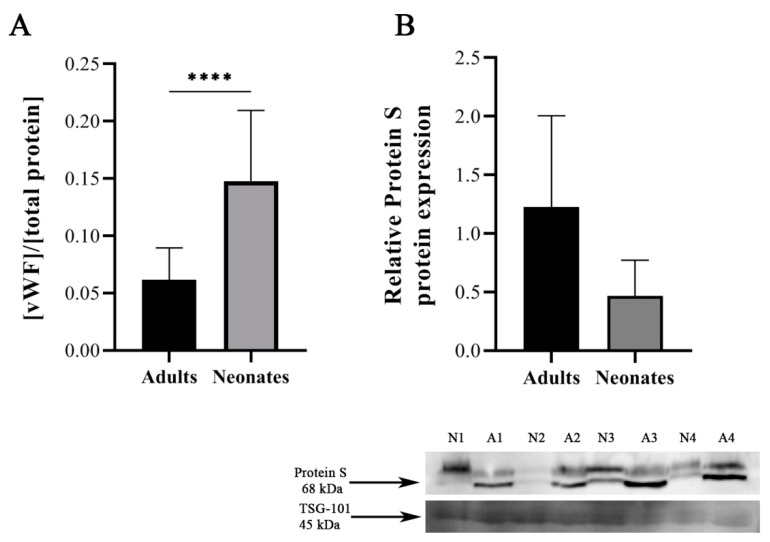
Validation of differential protein expression in independent samples. (**A**) vWF levels detected by ELISA in exosomes of adults (*n* = 10) and neonates (*n* = 10). ELISA analysis reveals statistically significant differences in von Willebrand factor (vWF) concentrations (Mann–Whitney U test). **** *p* < 0.0001. (**B**) Western blot of protein S in plasmatic exosomes from adults (A; *n* = 4) and neonates (N; *n* = 4) (unpaired *t*-test, one-tailed). Tumor susceptibility gene 101 (TSG-101) was evaluated as exosome-specific marker.

**Table 1 ijms-22-01926-t001:** Differential protein expression identified by sequential window acquisition of all theoretical mass spectrometry (SWATH MS) in plasma exosomes from neonates vs. adults (*p* < 0.05; FCh > 2 or <0.5).

Uniprot Code	Gene Name	Protein Name	*p*-Value	FCh
P02786	TFR1	Transferrin receptor protein 1	0.00104	15.32
Q99808	S29A1	Equilibrative nucleoside transporter 1	0.01424	14.01
P11166	GTR1	Solute carrier family 2, facilitated glucose transporter member 1	0.00106	10.15
P69892	HBG2	Hemoglobin subunit γ-2	0	9.43
P02792	FRIL	Ferritin light chain	0.00438	9.22
Q16819	MEP1A	Meprin A subunit α	0.01481	7.13
P02776	PLF4	Platelet factor 4	0.00809	7.01
Q02094	RHAG	Ammonium transporter Rh type A	0.00772	6.71
P27918	PROP	Properdin	0	6.26
P16671	CD36	Platelet glycoprotein 4	0.00001	5.99
Q9NQ84	GPC5C	G-protein-coupled receptor family C group 5 member C	0.00467	5.87
P08195	4F2	4F2 cell surface antigen heavy chain	0.00224	5.77
Q658P3	STEA3	Metalloreductase STEAP3	0.00262	5.34
Q6UX06	OLFM4	Olfactomedin-4	0.00008	5.08
P02787	TRFE	Serotransferrin	0.00101	5.05
P08514	ITA2B	Integrin αIIb	0	4.88
P02730	B3AT	Band 3 anion transport protein	0.00334	4.71
P11142	HSP7C	Heat shock cognate 71 kDa protein	0.00239	4.7
Q15758	AAAT	Neutral amino acid transporter B(0)	0.00829	4.56
P05164	PERM	Myeloperoxidase	0.00003	4.46
P0DMV9	HS71B	Heat shock 70 kDa protein 1B	0.0012	4.36
P27105	STOM	Erythrocyte band 7 integral membrane protein	0.00326	4.18
P02743	SAMP	Serum amyloid P-component	0	4.15
Q13201	MMRN1	Multimerin-1	0.00228	4.1
Q15907	RB11B	Ras-related protein Rab-11B	0.00058	4.03
Q9HD89	RETN	Resistin	0.00002	3.79
P01024	CO3	Complement C3	0	3.6
Q86UX7	URP2	Fermitin family homolog 3	0	3.51
P05023	AT1A1	Sodium/potassium-transporting ATPase subunit α-1	0.00875	3.39
P18428	LBP	Lipopolysaccharide-binding protein	0.00971	3.31
P04275	VWF	von Willebrand factor	0	3.29
Q00610	CLH1	Clathrin heavy chain 1	0.00667	3.15
P01859	IGHG2	Ig heavy constant γ 2	0.0001	3.06
Q9Y490	TLN1	Talin-1	0.00002	3.02
P00451	FA8	Coagulation factor VIII	0.00055	3.01
P05106	ITB3	Integrin β3	0.00003	3
P01042	KNG1	Kininogen-1	0.00652	2.95
O15400	STX7	Syntaxin-7	0.00007	2.89
P12259	FA5	Coagulation factor V	0	2.77
P26038	MOES	Moesin	0.00065	2.74
P10643	CO7	Complement component C7	0	2.73
Q9UP52	TFR2	Transferrin receptor protein 2	0.03685	2.71
P02760	AMBP	Protein AMBP	0.00118	2.7
P11597	CETP	Cholesterylester transfer protein	0.00012	2.7
P15144	AMPN	Aminopeptidase N	0.00896	2.6
P21926	CD9	CD9 antigen	0.00006	2.59
P01861	IGHG4	Ig heavy constant γ 4	0.00009	2.58
P0C0L5	CO4B	Complement C4-B	0	2.54
Q71DI3	H32	Histone H3.2	0.0305	2.49
P05546	HEP2	Heparin cofactor 2	0.00001	2.44
P08603	CFAH	Complement factor H	0.0057	2.43
P13671	CO6	Complement component C6	0.00044	2.33
P04004	VTNC	Vitronectin	0	2.3
Q9HAV0	GBB4	Guanine nucleotide-binding protein subunit β-4	0.00045	2.27
P21333	FLNA	Filamin-A	0.00019	2.23
P02788	TRFL	Lactotransferrin	0.01132	2.23
P62834	RAP1A	Ras-related protein Rap-1A	0.00141	2.2
P01717	LV325	Ig λ variable 3-25	0.04177	2.18
P02675	FIBB	Fibrinogen β chain	0.00005	2.16
P69905	HBA	Hemoglobin subunit α	0.00317	2.15
P63104	1433Z	14-3-3 protein ζ/δ	0.00005	2.14
P02679	FIBG	Fibrinogen γ chain	0	2.13
P01857	IGHG1	Ig heavy constant γ 1	0.00004	2.1
P01023	A2MG	A-2-macroglobulin	0.00104	2.02
P06312	KV401	Ig κ variable 4–1	0.00234	0.48
P06310	KV230	Ig κ variable 2–30	0.00695	0.46
P07225	PROS	Vitamin K-dependent protein S	0.0011	0.42
P49721	PSB2	Proteasome subunit β type-2	0.01941	0.4
P01834	IGKC	Ig κ constant	0.00001	0.4
P01766	HV313	Ig heavy variable 3–13	0.00043	0.4
P0DOY3	IGLC3	Ig λ constant 3	0	0.39
A0A0C4DH69	KV109	Ig κ variable 1–9	0.00017	0.39
P04430	KV116	Ig κ variable 1–16	0.00037	0.37
B9A064	IGLL5	Ig λ-like polypeptide 5	0.00002	0.36
P27169	PON1	Serum paraoxonase/arylesterase 1	0.00304	0.36
Q93050	VPP1	V-type proton ATPase 116 kDa subunit a isoform 1	0.01287	0.35
P01700	LV147	Ig λ variable 1–47	0	0.35
P00338	LDHA	L-lactate dehydrogenase A chain	0.01646	0.34
P04003	C4BPA	C4b-binding protein α chain	0	0.33
P01601	KVD16	Ig κ variable 1D–16	0.00931	0.32
P02766	TTHY	Transthyretin	0.01393	0.32
Q6NSI8	K1841	Uncharacterized protein KIAA1841	0.04514	0.31
Q16851	UGPA	UTP-glucose-1-phosphate uridylyltransferase	0.04304	0.31
A0A0B4J1X5	HV374	Ig heavy variable 3–74	0.00052	0.31
A0A0C4DH25	KVD20	Ig κ variable 3D–20	0	0.31
P0DP03	HV335	Ig heavy variable 3–30–5	0	0.3
A0A0C4DH24	KV621	Ig κ variable 6–21	0.00154	0.28
P08519	APOA	Apolipoprotein(a)	0.00237	0.28
A2NJV5	KV229	Ig κ variable 2–29	0.00394	0.27
A0A0A0MS15	HV349	Ig heavy variable 3–49	0	0.27
A0A0C4DH68	KV224	Ig κ variable 2–24	0.00088	0.26
P01624	KV315	Ig κ variable 3–15	0.00002	0.25
P28072	PSB6	Proteasome subunit β type-6	0.02057	0.25
P04433	KV311	Ig κ variable 3–11	0.00002	0.24
P01591	IGJ	Ig J chain	0.00001	0.23
O43866	CD5L	CD5 antigen-like	0.00001	0.2
A0A0B4J1Y9	HV372	Ig heavy variable 3–72	0.00097	0.17
A0A075B6H9	LV469	Ig λ variable 4–69	0.00042	0.15
P00738	HPT	Haptoglobin	0.00134	0.13
P00739	HPTR	Haptoglobin-related protein	0	0.11
O14791	APOL1	Apolipoprotein L1	0.00002	0.08
P01877	IGHA2	Ig heavy constant α 2	0.00001	0.05
Q8TF72	SHRM3	Protein Shroom 3	0.01003	0.05
P01876	IGHA1	Ig heavy constant α 1	0.00001	0.02

Ig, Immunoglobulin.

**Table 2 ijms-22-01926-t002:** Interaction analysis for proteins differentially expressed between neonate and adult exosomes.

**Upregulated in Neonates vs. Adults**			
**Pathway**	**Protein Count**	***p* Value**	**Proteins**
Hemostasis	23	<0.001	RAP1A; YWHAZ; KNG1; FLNA; GNAI2; ITGA2B; ITGB3; A2MG; F5; PF4; PFN1; HIST2H3A; TLN1; FGA; FGB; F8; HIST2H3C; HIST2H3D; TF; FGG; CD36; vWF; HBG2
Formation of fibrin clot (clotting cascade)	9	<0.001	KNG1; A2MG; F5; PF4; FGA; FGB; F8; vWF; FGG;
Integrin cell surface interactions	9	<0.001	RAP1A; ITGA2B; ITGB3; THBS1; VTN; TLN1; FGB; FGG; FGA;
Complement cascade	6	<0.001	MASP1; C3; C1R; C7; C6; C4B;
β3 integrin cell surface interactions	7	<0.001	ITGA2B; ITGB3; THBS1; VTN; FGA; FGB; FGG;
Common pathway	5	<0.001	F5; PF4; FGA; FGB; FGG;
Platelet activation, signaling and aggregation	10	<0.001	RAP1A; YWHAZ; FLNA; GNAI2; ITGA2B; ITGB3; PFN1; vWF; TF; TLN1;
Ephrin B-EPHB pathway	7	=0.002	RAP1A; ITGA2B; ITGB3; FGA; FGB; TF; FGG;
p130Cas linkage to MAPK signaling for integrins	4	=0.002	RAP1A; ITGA2B; ITGB3; TLN1;
GRB2: SOS provides linkage to MAPK signaling for integrins	4	=0.002	RAP1A; ITGA2B; ITGB3; TLN1;
β2 integrin cell surface interactions	5	=0.008	KNG1; FGA; FGB; C3; FGG;
Ephrin B reverse signaling	5	=0.011	ITGA2B; ITGB3; FGA; FGB; FGG;
Initial triggering of complement	4	=0.014	MASP1; C3; C1R; C4B;
Netrin-mediated repulsion signals	4	=0.014	RAP1A; ITGA2B; ITGB3; TLN1;
Recycling pathway of L1	5	=0.016	RAP1A; CLTC; ITGA2B; ITGB3; TLN1;
Proteoglycan syndecan-mediated signaling events	27	=0.025	RAP1A; YWHAZ; CLTC; KNG1; FLNA; HSPA1A; GNAI2; SLC3A2; HSPA8; CLU; ITGA2B; ITGB3; A2MG; THBS1; VTN; TLN1; FGA; FGB; HSPA1B; TFRC; TF; FGG; SLC2A1; SDCBP; LBP; STEAP3; HBG2
Intrinsic pathway	4	=0.03	KNG1; A2MG; F8; vWF;
Integrin αIIbβ3 signaling	4	=0.047	RAP1A; ITGA2B; ITGB3; TLN1;
**Downregulated in Neonates vs. Adults**			
**Pathway**	**Protein Count**	***p* value**	**Proteins**
Chylomicron-mediated lipid transport	3	=0.006	APOA1; APOB; APOC3;
Lipoprotein metabolism	3	=0.033	APOA1; APOB; APOC3;

## Data Availability

The data presented in this study are available in this article or in Appendix A.

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
