# Peer review of "Qualitative and Quantitative Comparison of Plasma Exosomes from Neonates and Adults"

_ijms, 2021, doi:10.3390/ijms22041926_

Round 1
Reviewer 1 Report
This is the first and elegant study investigating plasma-derived exosomes between neonates and adults. The authors did a good job explaining the objective and discussing results; however, some revisions are needed in order to improve the quality of the manuscript.
Introduction:
-The manuscript can benefit from referencing a variety of primary articles, other than review articles, when discussing differences between neonate and adult platelets.
-The authors should also mention differences in dense granule number and/or content between neonates and adults (Urban et al. Haematologica 2017. PMID: 27810994).
-The authors can consider mentioning differences between cord platelets vs. peripheral neonatal platelets in the introduction as well.
Results:
-Figure 3B: impossible to read the graph axes
-Page 7 of 21, line 173: Rap1 signaling has been shown to be important for hemostasis/thrombosis but dispensable for vascular integrity during development & inflammation (Stefanini et al. Blood 2018. PMID: 30131434). The author might consider Rap-1A as part of the first protein group (hemostasis/plt activation/coagulation) rather than third group (human development).
-Page 7 of 21, line 173: Also, I am not convinced that 4F2 cell-surface antigen (CD98) is expressed in platelets (PMID:18625289, 22460579). Since we're focusing mostly on platelet-derived exosomes, the author should consider NOT having a third group of proteins important for human development altogether.
-Table 2: the headings in this table seems to be opposite from claims made in the text. For example, proteins involved in hemostasis, clotting, complement, etc. should be "Up-regulated in neonates vs. adults" rather than "Down-regulated" as currently labeled. Vice versa, lipid transport and metabolism proteins should have been labeled as "Down-regulated in neonates vs. adults". Am I correct?
Minor:
-Page 13, line 371: "Third, we determined the protein..." Did the author mean "protein content"?
-Please double check spacing throughout the manuscript. Lots of missing spaces between 2 words.
Author Response
This is the first and elegant study investigating plasma-derived exosomes between neonates and adults. The authors did a good job explaining the objective and discussing results; however, some revisions are needed in order to improve the quality of the manuscript.
We would like to thank the reviewer for the kindly words and for giving us the opportunity to improve our research with his/her suggestions.
Introduction:
-The manuscript can benefit from referencing a variety of primary articles, other than review articles, when discussing differences between neonate and adult platelets.
We appreciate this suggestion from the reviewer and have incorporated the following new references into the revised version of the manuscript (Pages 2 and 18)
- Israels SJ, Cheang T, Roberston C, McMillan-Ward EM, McNicol A: Impaired signal transduction in neonatal platelets. Pediatr Res 1999;45:687-691.
- Schlagenhauf A, Schweintzger S, Birner-Gruenberger R, Leschnik B, Muntean W: Newborn platelets: lower levels of protease-activated receptors cause hypoaggregability to thrombin. Platelets 2010;21:641-647.
- Corby DG, O'Barr TP: Decreased alpha-adrenergic receptors in newborn platelets: cause of abnormal response to epinephrine. Dev Pharmacol Ther 1981;2:215-225.
- Gelman B, Setty BN, Chen D, Amin-Hanjani S, Stuart MJ: Impaired mobilization of intracellular calcium in neonatal platelets. Pediatr Res 1996;39:692-696.
- Baker-Groberg SM, Lattimore S, Recht M, McCarty OJ, Haley KM: Assessment of neonatal platelet adhesion, activation, and aggregation. J Thromb Haemost 2016;14:815-827.
-The authors should also mention differences in dense granule number and/or content between neonates and adults (Urban et al. Haematologica 2017. PMID: 27810994).
We appreciate the reviewer’s observation and the text (along with the new reference) was modified accordingly (page 2, 4th paragraph, lines 76-78).
-The authors can consider mentioning differences between cord platelets vs. peripheral neonatal platelets in the introduction as well.
We appreciate the reviewer’s comment. The new statement along with the new references can be found at the first paragraph of the introduction.
Results:
-Figure 3B: impossible to read the graph axes
As suggested by the reviewer, the axes have been modified for a better identification and understanding (Page 8).
-Page 7 of 21, line 173: Rap1 signaling has been shown to be important for hemostasis/thrombosis but dispensable for vascular integrity during development & inflammation (Stefanini et al. Blood 2018. PMID: 30131434). The author might consider Rap-1A as part of the first protein group (hemostasis/plt activation/coagulation) rather than third group (human development).
-Page 7 of 21, line 173: Also, I am not convinced that 4F2 cell-surface antigen (CD98) is expressed in platelets (PMID:18625289, 22460579). Since we're focusing mostly on platelet-derived exosomes, the author should consider NOT having a third group of proteins important for human development altogether.
We appreciate the reviewer’s comments and, as suggested, we have considered just two groups in which up-regulated proteins in plasma exosomes from neonates fall into. The protein 4F2 cell-surface antigen has also been removed from the first group description (Page 7, lines 174-183).
-Table 2: the headings in this table seems to be opposite from claims made in the text. For example, proteins involved in hemostasis, clotting, complement, etc. should be "Up-regulated in neonates vs. adults" rather than "Down-regulated" as currently labeled. Vice versa, lipid transport and metabolism proteins should have been labeled as "Down-regulated in neonates vs. adults". Am I correct?
The reviewer is right and the mistake has been corrected accordingly (Page 9).
Minor:
-Page 13, line 371: "Third, we determined the protein..." Did the author mean "protein content"?
The reviewer is right. The misspelling has been amended (Page 13, line 375).
-Please double check spacing throughout the manuscript. Lots of missing spaces between 2 words.
We apologize for this mistake. The text has been revised as suggested and missing spaces have been corrected.
Reviewer 2 Report
The manuscript is elegantly written and very well organized.
The central topic is of fundamental importance to better investigate exosomal diversity, since exosomes currently have the potential to become the fundamental approach of liquid biopsy especially in the tumor field (PMID: 32759810) and the need to investigate exosomal diversity starting from the first years of life of patients - at an early stage - it seems to me an excellent approach to lay the foundations for more precise analyzes.
Moreover, in line 370 has been raised the problem of the non-existence, to date, of a standardized method that is effective for isolating exosomes without errors; something that the committee for the standardization and rigor of EVs studies is trying to put on paper
I suggest improving the resolution and quality of figure 4. I also noticed several missing spaces between words and small spelling errors.
Author Response
Answers to reviewer 2
The manuscript is elegantly written and very well organized.
We would like to thank the reviewer for the kindly words and for giving us the opportunity to improve our research with his/her suggestions.
The central topic is of fundamental importance to better investigate exosomal diversity, since exosomes currently have the potential to become the fundamental approach of liquid biopsy especially in the tumor field (PMID: 32759810) and the need to investigate exosomal diversity starting from the first years of life of patients - at an early stage - it seems to me an excellent approach to lay the foundations for more precise analyzes.
We really appreciate the comment of the reviewer. Accordingly, we have incorporated these sentences in the manuscript (Page 14, lines 382-386).
Moreover, in line 370 has been raised the problem of the non-existence, to date, of a standardized method that is effective for isolating exosomes without errors; something that the committee for the standardization and rigor of EVs studies is trying to put on paper.
We agree with the reviewer's comment and this reflection has also been incorporated into the manuscript (Page 13, lines 372-375) together with a new reference (48).
I suggest improving the resolution and quality of figure 4. I also noticed several missing spaces between words and small spelling errors.
As suggested by the reviewer, the quality of figure 4 and the missing spaces and spelling errors have been amended
Round 2
Reviewer 1 Report
I thank the authors for addressing all my comments/ suggestions. I have no further comments.